# Comparison of indoor temperatures of homes with recommended temperatures and effects of disability and age: an observational, cross-sectional study

Gesche M Huebner,[1] Ian Hamilton,[1] Zaid Chalabi,[2] David Shipworth,[1] Tadj Oreszczyn[1]

## ABSTRACT

**Objectives** We examine if temperatures in winter in English homes meet the recommendation of being at least 18°C at all times. We analyse how many days meet this criterion and calculate the hours per day and night being at/above 18°C. These metrics are compared between households with occupants aged above 64 years or having a long-term disability (LTD) and those younger and without disability.

**Design** Cross-sectional, observational.

**Setting** England.

**Participants** 635 households.

**Outcomes measures** (1) Mean temperatures, (2) proportion of days of the measurement period meeting the criterion, (3) average hours at/above 18°C, (4) average hours at night at/above 18°C.

**Results** Mean winter temperatures in the bedroom were $M_{BR}$=18.15°C (SD=2.51), the living room $M_{LR}$=18.90°C (SD=2.46) and the hallway $M_{Hall}$=18.25°C (SD=2.57). The median number of days meeting the criterion was 19–31%. For the living room, more days meet the criterion in the group with a LTD ($M_{disability}$=342 vs $M_{no\_disability}$=301, 95% CI 8 to 74), and with someone over 64 years present ($M_{above64}$=341, $M_{below65}$=301 95%, CI 8 to 74). The median number of hours/day meeting the criterion was 13–17. In the living room, households with a disability had more hours at 18°C ($M_{disability}$=364, $M_{no\_disability}$=297, 95% CI 17 to 83) as did the older age group ($M_{above64}$=347, $M_{below65}$=296, 95% CI 18 to 84). In the hallway, more hours met the criterion in households with a disability ($M_{disability}$=338, $M_{no\_disability}$=302, 95% CI 3 to 70). 247 homes had at least nine hours of at least 18°C at night; no effect of age or disability.

**Conclusions** Many households are at risk of negative health outcomes because of temperatures below recommendations.

[1]UCL Energy Institute, University College London, London, UK
[2]Department of Social and Environmental Health Research, London School of Hygiene and Tropical Medicine, London, UK

**Correspondence to**
Dr Gesche M Huebner;
g.huebner@ucl.ac.uk

## Strengths and limitations of the study

► This is the first analysis that allows examining the specific objective of comparing empirical temperature measurements to recommendations.
► The data set used corresponds to a largely representative sample in England.
► Despite outlier correction, it is possible that days were retained in the data set in which the dwelling was empty.
► Only three rooms in the homes were monitored as opposed to every room in a house. Some rooms that were monitored may not have been occupied.
► All three winter months were relatively mild; it is likely that colder winters would mean an even greater discrepancy between recommendations and realised temperatures.

in homes and health concluded that results from the retrieved studies were sufficient to recommend a temperature of at least 18°C for the whole population at all times.[2] The 18°C threshold was judged particularly important for people over 65 years or with pre-existing medical conditions, with a particular emphasis on it being kept at night.

The need for an indoor temperature threshold arises from the burden of excess winter mortality in England; 15% more deaths occur in winter months than non-winter months, corresponding to about 24 000 extra deaths per winter,[3] significantly higher than in other European countries.[4] While a net of complex factors impacts on winter deaths, the poor state of housing and health inequalities are major reasons for the higher excess winter death rate in England.[2 5] Excess winter deaths increase significantly with age of occupants, age of the property and poorer thermal efficiency ratings and are associated with lower indoor temperatures.[5 6] A meta-analysis on the effects of implementing energy efficiency

## INTRODUCTION

The 2016 Cold Weather Plan for England recommended 18°C as day and night minimum temperature for those 65 and older or anyone with pre-existing medical conditions;[1] and a recent systematic review on the link between internal temperatures

**BMJ**

measures that generally make it easier and more affordable to keep homes warm showed that there is a small but significant positive effect on health.[7]

This paper examines to what extent homes in England meet temperature recommendations in winter by comparing empirical data from 635 homes to the recommendation of 18°C as suggested.[2]

Temperatures vary widely between homes and over the course of a day.[8] Average temperatures during the heating season in England were 19.3°C for the living room, 18.8°C for the hallway and 18.9°C for the bedroom, based on the Energy Follow-Up survey.[9] While these indoor temperatures are above the recommended 18°C, they reflect the average across homes and days. Given the known variability between homes, a substantial number of homes likely had temperatures below the recommendation. Analysis of indoor temperature during cold conditions have shown considerable variability in temperatures among older households that is modified by dwelling energy performance and socioeconomic conditions.[10]

To our knowledge, no study has assessed to what extent homes in England meet the recommended temperatures. The objectives of this paper are to investigate:

▶ Number of days in winter meeting the criterion.
▶ Average number of hours per day meeting the criterion.
▶ Average number of hours per night meeting the criterion.
▶ Comparison of the above metrics depending on whether someone in the household has a long-term disability (LTD) or is over 64 years.

This paper does not aim at explaining reasons behind the differences found, for example, whether they are due to housing factors, income, personal choice and so on but focuses on assessing the 'status quo' that is, situation as measured in the study.

## METHODS
### Data
This study used data from the 2011 Energy Follow Survey (EFUS) commissioned by the Department of Business, Energy and Industrial Strategy (BEIS) (then the Department of Energy and Climate Change),[11] a large-scale cross-sectional national survey in England, and its parent-survey, the English Housing Survey (EHS), a national survey of people's housing circumstances, characteristics and condition. The EFUS survey consisted of an interview survey of a subset of households (n=2616) that had been first visited as part of the 2010/2011 EHS. A subset of those interviewed (n=943) consented to having temperature loggers in up to three rooms of the house set to record temperatures every 20 min from February 2011 to January 2012. It is not known if there was any systematic difference in temperature between those who consented to loggers and those who did not, but this is unlikely given that the households with loggers were broadly representative in regards to Census data (see table 1).

**Table 1** Comparison of sample characteristics to 2011 Census data

| | N in sample | % in sample | % in 2011 Census |
|---|---|---|---|
| Region[23] | | | |
| North East | 44 | 6.93 | 4.90 |
| North West | 103 | 16.22 | 13.30 |
| Yorkshire & Humber | 83 | 13.07 | 9.97 |
| East Midlands | 53 | 8.35 | 8.55 |
| West Midlands | 58 | 9.13 | 10.57 |
| East | 88 | 13.86 | 11.03 |
| London | 46 | 7.24 | 15.42 |
| South East | 101 | 15.91 | 16.29 |
| South West | 59 | 9.29 | 9.98 |
| Dwelling type[24] | | | |
| Detached | 153 | 24.09 | 22.30 |
| Semi-detached | 204 | 32.13 | 30.70 |
| Terraced (including end-terrace) | 178 | 28.03 | 24.50 |
| Purpose-built flats | 86 | 13.54 | 16.70 |
| Converted flat | 14 | 2.20 | 4.30 |
| In commercial building | 0 | 0.00 | 1.10 |
| Caravan, mobile home, etc | 0 | 0.00 | 0.40 |
| Tenure[25] | | | |
| Owned outright | 192 | 30.24 | 30.60 |
| Owned with a mortgage/loan | 211 | 33.23 | 32.80 |
| Shared ownership | na | na | 0.80 |
| Rented from council (Local Authority) | 82 | 12.91 | 9.40 |
| Social rented: Other | 94 | 14.80 | 8.30 |
| Private rented | 56 | 8.82 | 16.80 |
| Living rent free | na | na | 1.30 |

The linked data sets were explicitly made available by BEIS for this research project. Parts of the data set used in this study remain private (ie, the high-resolution temperature data and the connection identifier between the EFUS and EHS). The non-linked data sets and summarised temperature data are accessible via the UK Data Archive. As this paper constitutes secondary data analysis, no ethical approval was required and no personal data (ie, identifying individuals) was available or used.

Valid temperature data were obtained from n=823 households (see BRE[11] for details). For this paper, only those n=760 households with three rooms temperature monitored (bedroom, living room, hallway) were included. A total of 105 households were excluded because of changes to the household or home since the last EHS. Hence, the final sample size on which all

analyses are based is n=635 homes with approximately national representativeness on geographical location, tenure and dwelling type (see table 1).

Tenure is the only variable showing some larger discrepancy between the sample and census, with 'social rented: other' over-represented by about 7% and 'privately rented' under-represented by about 8%. Given that socially rented accommodation is generally the best in terms of energy efficiency and privately rented accommodation the worst, this mismatch might indicate that in a truly representative sample the criterion of at least 18°C would be met to a slightly lesser extent.

### Survey interview data for EHS and EFUS

Data were collected through computer-assisted personal interviewing in the home of the respondent. For the purpose of this study, only questions relating to age of the householder and their self-reported health were analysed.

Respondents were asked if they and other household members, where applicable, had any long-standing physical or mental health condition. If the question condition was affirmed, the interviewer asked for a specification (table 2).

Of the n=635 households, n=369 reported one or more LTD. For the purpose of this study, only the dichotomised variable of 'any LTD' versus 'no LTD' was used, irrespective of the type of condition and total number of individuals with LTDs in one household. Any LTD indicates vulnerability in the household and an adaptation of the environment would be required.

The second variable of interest was age of the oldest household member; age was dichotomised into '64 years and younger' and '65 years and older', because 65 years was the cut-off used for the specific recommendations on indoor temperatures.[1] Among the n=635 households, n=206 dwellings had the oldest household member of age 65 or older.

**Table 2** Prevalence of LTDs in the sample

| LTD type | Number of households in which prevalent |
| --- | --- |
| Vision | 28 |
| Hearing | 24 |
| Learning | 12 |
| Heart | 75 |
| Breathing | 92 |
| Mobility | 146 |
| Mental | 35 |
| Other | 165 |
| Don't know | 4 |

Note that in some households multiple LTD existed, that is, the 581 occurrences listed here were distributed across 369 households. LTD, long-term disability.

### Temperature recordings

Temperatures were recorded every 20 min using modified TinyTag Transit 2 data loggers, that have an accuracy of +/-0.2°C and a resolution of 0.01°C.[11] The temperature loggers were usually installed by the interviewer at the end of the EFUS interview, on an internal wall, away from heat sources and direct sunlight, at a height accessible by the householder but out of reach of small children.[11]

Temperature recordings for February 2011, December 2011 and January 2012 were used, that is, those months considered as winter by the Office for National Statistics[12] for which temperature data were available.[13] Note, the specific months monitored were mild compared with historic years, with February 2011 being 1.7°C milder across the UK than the UK average 1981–2010 and December 2011 and January 2012 being both 1°C milder than the 1981–2010 average.[14] Internal temperatures are dependent on external temperatures, hence the temperatures during colder years will be significantly lower than presented here. For every dwelling, an extreme value correction was performed on the combined temperature data from the 3 months where any data point more than 1.5 IQRs below the first quartile was removed as extremely low temperatures might reflect absence from the home. The median numbers of extreme values removed were 13, 2 and 8, for bedroom, living room and hallway, respectively.

### Derived variables

Four outcome variables were constructed from the recorded temperature readings.

#### (a) Mean temperature for each room over the winter period

For each dwelling and room, the average temperature across the three winter months was calculated.

#### (b) Days with temperatures at or above 18°C

We calculated the number of days in which homes met the criterion of being at least 18°C continuously. While a strict interpretation of the recommendation would mean that 100% of all measurements need to be at 18°C or above (ie, all 72 measurements), we relaxed the assumption to 94.4% of all measurements (ie, 68 out of 72 measurement points). This is meant to take into account that brief drops in temperature are entirely plausible, for example, due to window or door opening.

For each home, on each day and in each room, we checked at each measurement point if the temperature was at least 18°C, with a 1 recorded if it was and a 0 if it was not. The values for each day were summed up and divided by the total number of measurements per day. If 68 measurements were at 18°C or above, then the resulting value would be 68/72=0.94, that is, 94.4%. We calculated the percentage of days for which the temperature measurements during the day had 94.4% of values at 18°C or above.

The percentage of days meeting the criterion is reported instead of the absolute number as some homes

did not have temperature data recordings for all 90 days (median was 86 days).

### (c) Hours at or above 18°C

For each home, on each day and in each room, we calculated the average number of hours for which the temperature was at least 18°C per a 24 hours period. We checked if consecutive measurements, that is 20 min segments, were both at least 18°C, where each day lasted from midnight to midnight the next day. This meant that 2 days (30 January 2012, 28 February 2011) were excluded from analysis as there was no subsequent day. For each home, we averaged the estimated daily temperature metrics across all days, separately for each room.

### (d) Hours at or above 18°C during night

We defined night-time as lasting from 20:00 hours to 08:00 hours next day to take into account that people sleep at different times and identified whether 20 min segments of temperature readings (ie, two consecutive measurements) were at 18°C or above within the 12 hours time window. As above, 2 days were excluded. For each home, we averaged the estimated nightly temperature metrics across all days. Only the bedroom was considered.

Hence, four outcome variables were derived from the raw data for each dwelling. The first three, average temperatures (a), proportion of days meeting the criterion (b) and hours meeting the criterion (c), were calculated separately for each room. The final outcome variable, hours meeting the criterion at night, was only calculated for the bedroom, assuming that that is where people slept.

### Statistical analysis

For the normally distributed variable 'mean temperature' (outcome variable a), a repeated measures analysis of variance (ANOVA) was used to test for differences between rooms, and a generalised linear model with the fixed factors age and disability status and their interaction to test if temperatures differed depending on those variables. Posthoc comparisons were Bonferroni adjusted.

The non-normally distributed outcome variables (b)–(d) were analysed using ANOVA on ranks[15] whereby data are transformed into ranks (averaged in the case of ties) over the entire data set and then a parametric ANOVA is applied to the ranks. The rank 1 was assigned to the lowest value, that is, to 0 days meeting the criterion; a higher mean rank value indicates more days meeting the criterion. The main effects of age and disability were tested and their interaction. The presence of an interaction effect is to be interpreted with greatest caution as the procedure is associated with an increase in Type 1 error (ie, claiming statistical significance where there is none, see eg, Higgins and Tashtoush[16]); however, if no interaction effect is found, it can be assumed that indeed, there is not one.

Additionally, for days at or above 18°C (outcome variable (b)), relative risk was calculated following[17] for the

rooms where disability or age had a significant effect to be able to easily articulate how much more likely those more vulnerable were to live at the criterion.

### Patient and public involvement

As this paper constitutes secondary data analysis, there was no involvement of patients or the public.

## RESULTS

### Mean temperature for each room over the winter period

Across all dwellings, mean temperatures in the bedroom were $M_{BR}$=18.15°C (SD=2.51), the living room $M_{LR}$=18.90°C (SD=2.46) and the hallway $M_{Hall}$=18.25°C (SD=2.57). A repeated measures ANOVA showed a main effect of room type, $F(2, 1268)$=58.41, $p<0.001$. Posthoc comparisons showed the living room was significantly warmer than the bedroom ($p<0.001$; mean difference: 0.75, 95% CI for difference: 0.94 to 0.57) and hallway ($p<0.001$; mean difference: 0.65; 95% CI for difference: 0.57 to 0.94) which did not differ significantly from each other.

Figure 1 shows the probability density function (PDF) of the mean temperatures for the three rooms, created using the R package 'sm'.[18] The PDF is best understood through the area underneath it. The area underneath the PDF of a continuous random variable between two values gives the probability that the random variable is between those values. The total area underneath the PDF over the whole range of values of the random variable is unity.

Figure 1 indicates a wide spread in mean temperatures. While the average temperature (across days and homes) in all three rooms is slightly above 18°C, in a substantial number of homes, it was below 18°C. In the case of the bedroom, 286 dwellings (45%) had an average

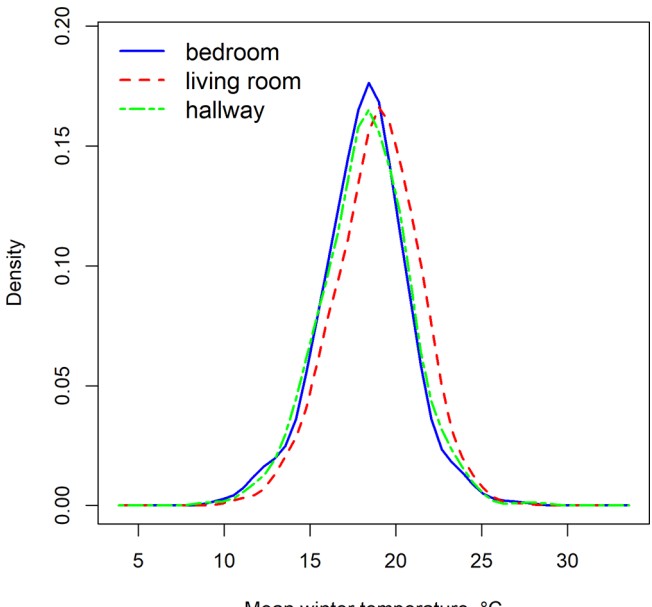

**Figure 1** Probability density function of mean winter temperatures in bedroom, living room and hallway.

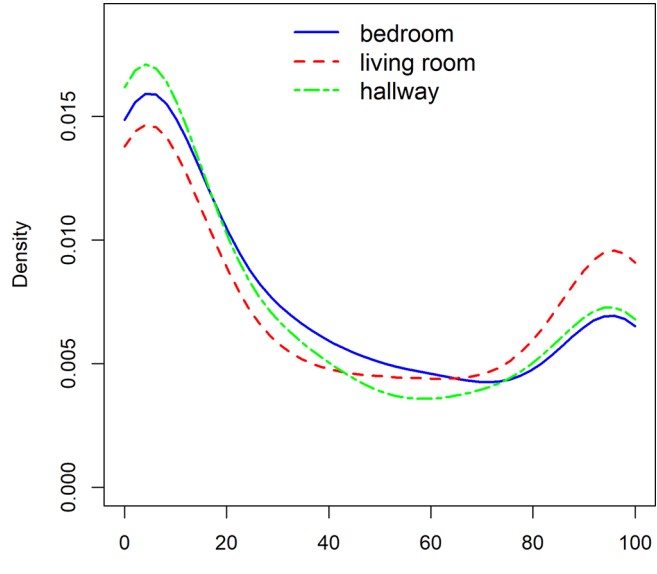

**Figure 2** Probability density function for proportion of days where 94.4% of days meet the criterion.

temperature below 18°C, in the living room 209 (33%) and in the hallway 278 dwellings (44%).

In the bedroom, only the effect of disability was significant (F(1, 631)=4.38, p=0.037) with higher temperatures in the group with disability ($M_{disability}$=18.35°C; $M_{no\_disability}$=17.87°C; 95% CI for difference: 0.03 to 0.94). For the hallway, again only the effect of disability was significant (F(1, 631)=7.64, p=0.006) with higher temperatures in the disability group ($M_{disability}$=18.58°C, $M_{no\_disability}$=17.93°C). There was a strong trend for higher temperatures in the homes of the older age group (p=0.059). In the living room, both the main effects of age (F(1, 631=12.39, p<0.001) and disability (F(1, 631)=15.53, p<0.001) were statistically significant. Temperature was higher in the group with disability ($M_{disability}$=19.37°C; $M_{no\_disability}$=18.50°C; 95% CI for difference: 0.44 to 1.30) and in the older age group ($M_{above64}$=19.32°C; $M_{below65}$=18.55°C; 95% CI for difference: 0.34 to 1.21).

### Days with temperatures above 18°C
We analysed the number of days during the winter on which dwellings met the indoor temperature criterion. Figure 2 shows the PDF of the distribution for the three rooms.

Figure 2 indicates that the largest share of homes do not meet the criterion but that a substantial number of homes

meet it on 90%–100% of days. For the bedroom, 11% of homes meet the criterion on all days and 17% on more than 90% of days. For the living room, the numbers are 15% and 24%, respectively, and for the hallway 12% and 17%. The median number of days that indoor temperatures meet the criterion on all days is $Md_{BR}$=22.6% of days, $MD_{LR}$=31.1% and $Md_{Hall}$=18.9%.

The ANOVA for ranks in the bedroom showed neither a main effect of age or disability nor an interaction effect. For the living room both the main effects of disability (F(1, 631)=6.00, p=0.015) and age (F(1, 631)=6.06, p=0.0114) were significant, with a higher share of days meeting the criterion in the group with a LTD ($M_{disability}$=342 vs $M_{no\_disability}$=301; 95% CI for difference: 8–74) and with someone over 64 years present ($M_{above64}$=341, $M_{below65}$=301; 95% CI for difference: 8–74). Of those households with LTD, 26.8% had a continuous temperature above 18°C on 90% of days or more compared with 20.7% for those without LTD. Expressed as a relative risk,[17] people with LTD are 1.30 times more likely to be living in dwellings where the temperature is consistently over 18°C compared with those without LTD, and people who are 65 years and above are 1.56 more likely than those below 65 years.

For the hallway, there were no significant effects. However, there was suggestive evidence of a trend towards more days meeting the criterion in the group with a LTD (p=0.064).

### Number of hours at which temperatures are at or above 18°C
The number of hours at or above 18°C were non-normally distributed with peaks at either extreme of 0 and 24 hours. The median number of hours at/above 18°C was $Md_{BR}$=14:01 hours per day, $Md_{LR}$=16:57 hours and $Md_{Hall}$=13:24 hours. Table 3 shows for how many hours in each room, depending on disability and age group, the criterion was met.

In the bedroom, there were no significant main or interaction effects. In the living room, both the main effect of disability (F(1, 631)=8.89, p=0.003) and of age (F(1, 631)=9.28, p=0.002) were significant, with more hours at or above 18°C in those households occupied by individuals with a disability ($M_{disability}$=364, $M_{no\_disability}$=297, 95% CI for difference: 17 to 83) and in the older age group ($M_{above 64}$=347, $M_{below 65}$=296, 95% CI for difference: 18 to 84). In the hallway, the main effect of disability was significant (F(1, 631)=4.53, p=0.034) and the effect of age approached significance (p=0.073) with again more hours meeting the criterion in the group with a

**Table 3** Median number of hours with temperatures at the criterion for the three rooms separated by disability and age group

| | Disability status | | Age | |
|---|---|---|---|---|
| | **No LTD** | **LTD** | **Below 65 years** | **Above 64 years** |
| Bedroom | 13:10 hours | 14:05 hours | 12:56 hours | 15:09 hours |
| Living room | 15:13 hours | 17:59 hours | 15:37 hours | 20:01 hours |
| Hallway | 10:58 hours | 14:52 hours | 12:35 hours | 16:01 hours |

LTD, long-term disability.

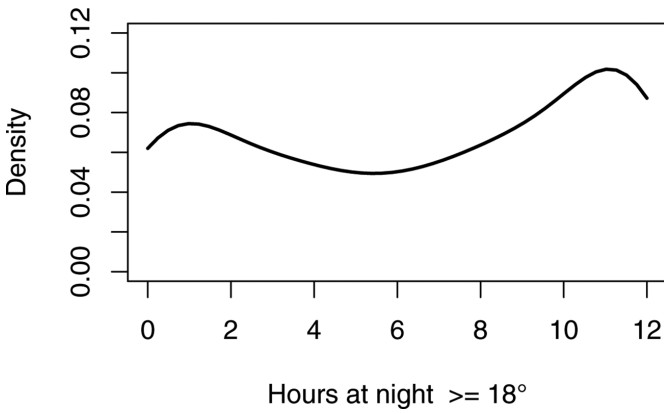

**Figure 3** Probability density function showing how many hours at night are at least at 18°C.

LTD ($M_{disability}$=338, $M_{no\_disability}$=302, 95% CI for difference: 3 to 70) and the older group.

### Night temperatures

We subsequently tested if the recommendation of 9 hours of 18°C at night time was met (see figure 3).

Across the full sample, 247 homes (38.9%) had at least 9 hours of temperatures of at least 18°C at night and 101 (15.9%) homes had less than 1 hour at the criterion. The median was 7:08 hours.

An ANOVA on ranks showed neither a main effect of disability nor of age and no interaction.

### DISCUSSION

This study is the first to establish whether measured temperature data in homes corresponds to the recommended temperatures, in the general population and the subgroups of those with a LTD and/or aged 65 and above. While average temperatures across homes in this sample are slightly above 18°C, the wide variability means that many homes have lower temperatures. Depending on room type, the recommended indoor temperature of 18°C was only met on 19%–31% of days during the three studied mild winter months. Only 5%–9% of homes met the criterion of being at least 18°C throughout the day, with up to 22% of homes meeting the criterion if set as having the recommended temperature throughout the day for at least 90% of days. Those with a disability and old age were 1.30 and 1.56 more likely to meet the condition than those without disability and the younger age group. The median number of hours per day at or above 18°C was 17 for the living room and 14 and 13½, respectively, for bedroom and hallway. The median number of hours meeting the criterion were one to 3 hours higher in households with a disability or aged 65 and above. At night, 37% of homes had temperatures of at least 9 hours at 18°C or more, with the median being 7 hours, with no effect of age group or disability.

In summary, the majority of measures employed showed that the recommendation was not met, neither in the overall sample nor within the subsamples of those more vulnerable to effects of cold.

### Limitations and strengths of this study

Despite outlier correction, it is possible that days were retained in the data set in which the dwelling was empty, leading to an underestimation of the criterion being met assuming that it only holds for occupied times. Only three rooms in the homes were monitored as opposed to every room in a house. The study is cross-sectional and cannot add evidence on whether low temperatures are associated with poor health outcomes. All three winter months were relatively mild (mean temperatures in February 2011 1.7°C above the 1981–2010 average; in December 2011 1.0°C above the 1981–2010 average; January 2012 1.0°C above the 1981–2010 average[14]); it is likely that colder winters would mean even lower prevalence of 18°C. Households consented to having temperature loggers installed; it is possible that temperatures in those households were either higher or lower than in those not giving consent.

This paper is the first analysis that allows examining the specific objective of comparing empirical temperature measurements to recommendations, showing a significant discrepancy and the need for action. The data set used corresponds to a largely representative sample in England; hence, results likely are generalisable to the whole of England.

### CONCLUSION

In summary, data showed that the majority of homes do not meet the recommendation, neither across the whole sample nor within the vulnerable subgroups. If living in homes below the temperature threshold is a determinant of cold-related ill health, then many English households are at risk of developing negative health outcomes. If this exposure presents a high risk to health, then substantial action is needed to increase temperatures in homes, be it through improvements in building fabric, extended use of heating systems or increased thermostat set points.

From an energy demand perspective, energy use in buildings would increase substantially when keeping all homes at 18°C continuously. Without improvement in the energy performance of buildings, for example, through fabric insulation and greater efficiency of heating systems, this outcome would result in an increase in heating energy use and move away from the UK's energy efficiency goals. Hence, implementing new and stricter policies on retrofitting are needed. The UK has been dubbed 'the cold man of Europe' given that in comparison to other European countries, it has one of the highest level of fuel poverty and some of the most inefficient housing stock, with 21 of out of 26 million dwellings rated as 'D' or below on their energy performance certificate.[19] Energy efficiency interventions have been shown to increase daytime living room temperatures by 1.6°C and night time bedroom temperatures by 2.8°C.[20] Increased energy efficiency can

bring the risk of higher temperatures in summer which might also be detrimental for health.[7]

There is also the question of whether individuals can afford to increase fuel expenditure to achieve the stated indoor temperature threshold. Mean energy expenditure was 4.4% of total household expenditure, with a substantially higher proportion of 9.7% in the lowest income decile.[21] Spending on fuel to increase temperatures would result in a greater proportion of household resources allocated to fuels. This increase in spending could result in a net cost-benefit if the health impacts were accounted for in these calculations[22]—but householders might not realise this directly.

**Contributors** GMH and IH conceived the study. GMH conducted the analysis. ZC advised on the analysis. GMH wrote the paper. IH, ZC, TO and DS proofread the manuscript multiple times.

**Funding** Funding for this research was provided by the Research Councils UK (RCUK) Centre for Energy Epidemiology, grant reference EP/K011839/1.

**Competing interests** None declared.

**Patient consent** Not required.

**Provenance and peer review** Not commissioned; externally peer reviewed.

**Data sharing statement** The linked data sets were explicitly made available by BEIS for this research project. Parts of the dataset used in this study remain private (ie, the high resolution temperature data and the connection identifier between the EFUS and EHS to link dwelling and household features). The individual, non-linked data sets and the summarised temperature data are available on UK Data Archives.

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
