## [Reviewer comments · BMJ Open]

ARTICLE DETAILS

TITLE (PROVISIONAL)	A comparison of indoor temperatures of homes to recommended temperatures and effects of disability and age – an observational, cross-sectional study
AUTHORS	Huebner, Gesche; Hamilton, Ian; Chalabi, Zaid; Shipworth, David; Oreszczyn, Tadj

VERSION 1 – REVIEW

REVIEWER	Christine Boomsma Leiden University, The Netherlands
REVIEW RETURNED	08-Jan-2018

GENERAL COMMENTS	The paper provides an analysis of indoor temperatures compared to the recommended temperature in the UK, specifically looking at the effects of age and disabilities. The objectives and analysis are for the most part clearly described. But some clarifications and/or changes are needed: Abstract p3. line 7. change to England p3. line 7-10. rephrase sentence, seems incomplete. Objectives are more clearly described on p5-6, maybe try to align the text in the abstract and introduction better. p3. line 25. "proportion of days" - clarify proportion, proportion of days out of the year/measurement period? Introduction Literature on the relationship between cold homes and ill health is missing. e.g.: Liddell, C., Morris, C., 2010. Fuel poverty and human health: a review of recent evidence. Energy Policy 38 (6), 2987–2997. Maidment, C.D., Jones, C.R., Webb, T.L., Hathway, E.A., Gilbertson, J.M., 2014. The impact of household energy efficiency measures on health: a meta-analysis. Energy Policy 65, 583–593. p5. line 35-37: please clarify what data set these averages are coming from. Method p9. line 40: shouldn't this be: temperatures at or above 18?
---

	p9. line 29-36. The description of the derived variables could use some clarification. As there are different levels of measurements (per day/per hour/per month), the text is difficult to follow at times. Perhaps a table summarising the main variables would be helpful. As I read it, the following variables are used in the research: 1) Average temperature for each room over the 3 month winter period. 2) Percentage of days during 3 month winter period for which the temperature measurements throughout the day had 94.4% of values at 18 C or above. 3) Average number of hours for which the temperature was at least 18 C per 24-hour period during the 3 month winter period. 4) Average number of hours for which the temperature was at least 18 C per 12-hour night time period during the 3 month winter period. Results p11. line 32. Some more explanation of what a 'probability density function' shows would be helpful for readers not familiar with this type of plot. p13. caption for Table 3 is included twice. Discussion p.14. line 22. I think this should be: 'or aged 65 or above'
--	--

REVIEWER	Ulla Haverinen-Shaughnessy Tampere University of Technology, Finland
REVIEW RETURNED	13-Jan-2018

GENERAL COMMENTS	The topic of this paper is very current and relevant from environmental health point of view. The paper is well written and the methodology has been described with clarity. Some strengths of the study include objective measurements and a large sample of English housing stock. Table 1 shows up to 8% differences between the sampled homes and census data, with largest differences are seen in the tenure status. However, the authors state that "the households with loggers were broadly representative in regards to Census data". I suggest adding some discussion about whether the observed differences could have any impacts on how generalizable the results are. It should also be relevant to report if the age of the respondents or occurrence of long-term disabilities are independent of the tenure status among the sampled households. If possible considering the sample size, it would also be interesting to see if the type of disability had any impact on the reported associations. The paper is focused on winter indoor temperatures in English housing stock. The measurement data collection lasted twelve months, so some consideration on whether the authors have or will look at the acceptability of summer temperatures could be added. Exposure to high indoor temperatures is also dependent on housing quality, and is associated with increased mortality. In addition, discussion about how the situation in England compares with other countries could be added. See for example recent studies from Finland on the impact of age and tenure status on thermal comfort: Pekkonen M, Turunen M, Haverinen-Shaughnessy U. Housing quality perceptions in Finland: the elderly population. Building Research & Information. 2017 May 16:1-3.
--

	Pekkonen M, Liulu D, Skön J-P, Raatikainen M, Haverinen-Shaughnessy U. The influence of tenure status on housing satisfaction and indoor environmental quality in Finnish apartment buildings, Building and Environment , published online March, 2015.
REVIEWER	Vanessa Powell-Hoyland Sheffield Hallam University, UK
REVIEW RETURNED	17-Jan-2018
GENERAL COMMENTS	General: This paper addresses a well know important and current Public Health issue. While it is a secondary analysis it is well-conducted. Title is clear and informs the reader. Discussion In my option the paper is well written, it confirms what we already know around the challenges of older people with long term health conditions: The authors appropriately cite past literature to strength the case however referencing the findings from the Warm Front Better Health evaluation would aid the case http://www4.shu.ac.uk/research/cresr/warm-front-better-health-health-impact-evaluation-warm-front-scheme-0. The findings of the length of time a person heats their home should be of interest to the reader; I am not aware of any studies that have undertaken this method and therefore brings in new information. As a qualitative researcher it would have been interesting to have asked the participants if they sleep in their living rooms and their level of income, this would have enhanced the findings. I would recommend that the second reviewer has statistical experience. Do the authors have any recommendations for practice or policy?

VERSION 1 – AUTHOR RESPONSE

We would like to thank all three reviewers for their very helpful comments which we have all addressed as outlined below.

There were a couple of suggestions on additional analyses and factors to investigate. Although we are in full agreement that they would be very interesting analyses to perform, we felt that they were beyond the scope of the current paper, given: (1) the specific aim of the paper which is to compare recommended to empirical temperatures, (2) the word limit on the article, and (3) the fact that the paper is already quite dense with analyses done on a number of derived variables. We hope our detailed response below to these suggestions is accepted. We also note that the paper's length has already increased somewhat after implementing other suggestions, and so we believe that any additional analyses should be part of a subsequent paper.

We very much appreciate the overall positive evaluation, and hope that our responses are satisfactory and that our manuscript will be acceptable.

We respond below (in italics) to the detailed comments of the reviewers (reproduced in normal font)

We would also like to thank the Editor for pointing out the need for another proofreading which we have carried out.

Best regards, Gesche (for all authors)

Reviewer: 1

Abstract

p3. line 7. change to England

This has been changed to 'English homes' to clarify the setting.

p3. line 7-10. rephrase sentence, seems incomplete. Objectives are more clearly described on p5-6, maybe try to align the text in the abstract and introduction better.

We have rephrased this paragraph, streamlining it with the objectives mentioned later. It now says: 'We analyse how many days meet this criterion, and calculate the hours per day and night being at/above 18°C. These metrics are compared between households with occupants aged above 64 years or having a long-term disability, and those younger and without disability.'

This lengthened the abstract beyond the permissible word count; hence, we shortened it at other parts (see tracked changes).

p3. line 25. "proportion of days" - clarify proportion, proportion of days out of the year/measurement period?

We have changed this as follows to indicate it is the of the measurement period: 'Proportion of days of the measurement period meeting the criterion.'

Introduction

Literature on the relationship between cold homes and ill health is missing.

e.g.:

Liddell, C., Morris, C., 2010. Fuel poverty and human health: a review of recent evidence.

Energy Policy 38 (6), 2987–2997.

Maidment, C.D., Jones, C.R., Webb, T.L., Hathway, E.A., Gilbertson, J.M., 2014. The impact of household energy efficiency measures on health: a meta-analysis. Energy Policy 65, 583–593.

We have deliberately not reviewed any primary evidence on the link between cold homes and health given the very recent systematic review on this topic (Jevons R, Carmichael C, Crossley A, Bone A. Minimum indoor temperature threshold recommendations for English homes in winter - A systematic review. Public Health. 2016;136:4–12.) that had reviewed all relevant studies. Hence, attempting to carry out a full systematic review would have been redundant and beyond the scope of this paper, and furthermore a limited review incorporating only a few studies would not be widely informative. Hence, we prefer to refer to the above mentioned review as providing the background rationale, namely that there is a need to set indoor temperature thresholds to avoid detrimental health effects. We are very grateful to the reviewer for pointing out the meta-analysis on impacts of energy efficiency measures on health and have included a statement on it. This now says (page 5): 'A meta-analysis on the effects of implementing energy efficiency measures that generally make it easier and more affordable to keep homes warm, showed that there is a small but significant positive effect on health (7).'

p5. line 35-37: please clarify what data set these averages are coming from.

We have added two explanatory phrases to this ('in England', 'Energy Follow-Up Survey').

Method

p9. line 40: shouldn't this be: temperatures at or above 18?

Thanks so much for spotting this, we have changed it to 'at or above'.

p9. line 29-36. The description of the derived variables could use some clarification. As there are different levels of measurements (per day/per hour/per month), the text is difficult to follow at times. Perhaps a table summarising the main variables would be helpful. As I read it, the following variables are used in the research: 1) Average temperature for each room over the 3 month winter period. 2) Percentage of days during 3 month winter period for which the temperature measurements throughout the day had 94.4% of values at 18 C or above. 3) Average number of hours for which the temperature was at least 18 C per 24-hour period during the 3 month winter period. 4) Average number of hours for which the temperature was at least 18 C per 12-hour night time period during the 3 month winter period.

We agree that a table summarizing the variables might be helpful but we are already exceeding the number of recommended figures / tables by 1, and so we would prefer not to add another table. The reviewer's understanding of the variables is perfectly accurate. We have added a sentence to summarise the variables which now says: 'Hence, four outcome variables were derived from the raw data for each dwelling. The first three, average temperatures (a), proportion of days

meeting the criterion (b), and number of hours meeting the criterion (c), were calculated separately for each room. The final outcome variable, hours meeting the criterion at night, was only calculated for the bedroom, assuming that that is where people slept.'

Results

p11. line 32. Some more explanation of what a 'probability density function' shows would be helpful for readers not familiar with this type of plot.

We have added a couple of explanatory sentences on this. It now says: 'The PDF is best understood through the area underneath it. The area underneath the PDF of a continuous random variable between two values gives the probability that the random variable is between those values. The total area underneath the PDF over the whole range of values of the random variable is unity.'

p13. caption for Table 3 in included twice.

The first 'caption' was actually meant to be the text referring to the table. However, it did indeed look like a caption, so we have rephrased and repositioned it.

Discussion

p.14. line 22. I think this should be: 'or aged 65 or above'
Yes, thank you, this is changed!

Reviewer: 2

Reviewer Name: Ulla Haverinen-Shaughnessy Institution and Country: Tampere University of Technology, Finland Competing Interests: None declared

The topic of this paper is very current and relevant from environmental health point of view. The paper is well written and the methodology has been described with clarity. Some strengths of the study include objective measurements and a large sample of English housing stock.

Table 1 shows up to 8% differences between the sampled homes and census data, with largest differences are seen in the tenure status. However, the authors state that "the households with loggers were broadly representative in regards to Census data". I suggest adding some discussion about whether the observed differences could have any impacts on how generalizable the results are. We have added a few sentences on the potential effect of the one larger mismatch between sample and census after the table. This now says: 'Tenure is the only variable showing some larger discrepancy between sample and census, with 'social rented: other' overrepresented by about 7% and 'privately rented' underrepresented by about 8%. Given that socially rented accommodation is generally the best in terms of energy efficiency, and privately rented accommodation the worst, this mismatch might indicate that in a truly representative sample the criterion of at least 18°C would be met to a slightly lesser extent.'

It should also be relevant to report if the age of the respondents or occurrence of long-term disabilities are independent of the tenure status among the sampled households. If possible considering the sample size, it would also be interesting to see if the type of disability had any impact on the reported associations.

We entirely agree that this would be interesting additional analyses. However, we would prefer not to include them in the current paper for three main reasons: (1) the overall aim is to check whether temperatures per se, across tenure or any other classification such as dwelling type, meet the criterion, (2) the paper is already borderline long (with 4000 words the recommended length, and it being a couple of hundreds words longer); hence, to introduce, perform, report and discuss the additional findings would mean that the paper would become even denser than what it is now and if we omit many details to reduce paper size this is likely to have detrimental effect on the paper's clarity, and (3) splitting up the temperature data based on type of disability would likely create too small sample sizes, in particular considering the comorbidity of individuals with long-term disability, and the need then to consider interactions. We will take this analyses into account for future work.

The paper is focused on winter indoor temperatures in English housing stock. The measurement data collection lasted twelve months, so some consideration on whether the authors

have or will look at the acceptability of summer temperatures could be added. Exposure to high indoor temperatures is also dependent on housing quality, and is associated with increased mortality. Again, we entirely agree that summer temperatures and overheating risks are an important topic, but we again feel that it is beyond the scope of this paper to include this analysis. The paper is already borderline long and very dense. We will consider overheating in future work. We have included a sentence in the discussion picking up on this point, this now says: 'Increased energy efficiency can bring the risk of higher temperatures in summer which might also be detrimental for health (7).'

In addition, discussion about how the situation in England compares with other countries could be added. See for example recent studies from Finland on the impact of age and tenure status on thermal comfort:

Pekkonen M, Turunen M, Haverinen-Shaughnessy U. Housing quality perceptions in Finland: the elderly population. *Building Research & Information*. 2017 May 16:1-3.

Pekkonen M, Liulu D, Skön J-P, Raatikainen M, Haverinen-Shaughnessy U. The influence of tenure status on housing satisfaction and indoor environmental quality in Finnish apartment buildings, *Building and Environment*, published online March, 2015. We believe that the above mentioned papers are beyond the scope of this paper which is to ascertain whether achieved temperatures in English homes meet recommended temperatures. We are not aware of publications that have done similar analyses in other countries. However, to give a somewhat broader perspective, we have included the following statement in the discussion: 'The UK has been dubbed 'the cold man of Europe' given that in comparison to other European countries, it has one of the highest level of fuel poverty and some of the most inefficient housing stock, with 21 of out of 26 million dwellings rated as 'D' or below on their energy performance certificate. '

Reviewer: 3

Reviewer Name: Vanessa Powell-Hoyland

Institution and Country: Sheffield Hallam University, UK Competing Interests: None

General: This paper addresses a well know important and current Public Health issue. While it is a secondary analysis it is well-conducted.

Title is clear and informs the reader.

Discussion

In my option the paper is well written, it confirms what we already know around the challenges of older people with long term health conditions: The authors appropriately cite past literature to strength the case however referencing the findings from the Warm Front Better Health evaluation would aid the case <http://www4.shu.ac.uk/research/cresr/warm-front-better-health-health-impact-evaluation-warm-front-scheme-0>.

Thank you for pointing out that we haven't included the Warm Front Scheme evaluation; we are now referring to a relevant paper that shows how it increased temperatures in homes (p. 16), showing that investment in energy efficiency measures is crucial: 'Energy efficiency interventions, have been shown to increase daytime living room temperatures by 1.6 °C, and night time bedroom temperatures by 2.8 °C (23).'

The findings of the length of time a person heats their home should be of interest to the reader; I am not aware of any studies that have undertaken this method and therefore brings in new information.

We understand by this comment that the reviewer is asking us to determine if length of heating duration impacts on the hours / days meeting the criterion. Whilst we agree that this is an interesting issue, we see it as beyond the scope of the paper to conduct a detailed analysis on determinants of the derived variables; given the constraint on word count, and the overall aim of the paper which is to on compare observed temperatures to recommended temperatures (see also next point).

As a qualitative researcher it would have been interesting to have asked the participants if they sleep in their living rooms and their level of income, this would have enhanced the findings. We agree that it would have been interesting to know if people sleep in living rooms; however, this question wasn't asked in the survey. A follow-up study should certainly look more at explanatory factors of the derived variables, e.g. what factors are associated with hours per day meeting the criterion for which income might well play a role. However, given the constraint on word count, and the already rather dense paper, there isn't scope for this in the current study.

I would recommend that the second reviewer has statistical experience.

We are confident that between the three reviewers and the Editor, there was sufficient statistical expertise to comment on the statistical aspects.

Do the authors have any recommendations for practice or policy?

We have included one phrase on policy, indicating that better / stricter policies are needed to ensure greater energy efficiency. 'Hence, implementing new and stricter policies on retrofitting are needed.' (page 16).

VERSION 2 – REVIEW

REVIEWER	Ulla Haverinen-Shaughnessy Tampere University of Technology, Finland
REVIEW RETURNED	07-Mar-2018
GENERAL COMMENTS	I accept the authors' responses to my comments and have no further comments